# Βarriers and Gaps to Medical Care for Transgender Individuals: A TRANSCARE Scoping Review with a Focus on Greece

**DOI:** 10.3390/healthcare12060647

**Published:** 2024-03-13

**Authors:** Marilena Anastasaki, Agapi Angelaki, Philippos Paganis, Elena Olga Christidi, Nancy Papathanasiou, Eleni Panagiota Stoupa, Evika Karamagioli, Emmanouil Pikoulis, Christos Lionis

**Affiliations:** 1Clinic of Social and Family Medicine, School of Medicine, University of Crete, 710 03 Heraklion, Greece; 2Orlando LGBT+, 11527 Athens, Greecenp@orlandolgbt.gr (N.P.); 3Post Graduate Programme of Global Health–Disaster Medicine, School of Medicine, National and Kapodistrian University of Athens, 11527 Athens, Greece

**Keywords:** transgender care, primary care, barriers to care

## Abstract

Trans people face substantial barriers to care worldwide. In Greece, limited evidence regarding trans health and actions to improve accessibility in healthcare services is available. This study aims to identify barriers to care for transgender populations in order to discuss the potential gaps in healthcare providers’ training in this direction. A scoping review was conducted in PubMed. Study eligibility criteria included: (a) reporting on at least one barrier to care for trans individuals or at least one educational need for healthcare providers; (b) free full text availability; and (c) publishing from 2015 and afterwards. Discrepancies in study inclusion were discussed between the research team until consensus was reached. Out of 560 identified references, 69 were included in this study, with only three reporting empirical research from Greece. Several individual-, interpersonal-, and institutional-level barriers to healthcare for trans individuals were identified. These included discriminatory treatment by healthcare providers, a lack of knowledgeable providers trained on trans-specific healthcare issues, lack of trusted and safe healthcare environments, health coverage-related issues, and healthcare systems that do not take into account particular transgender health issues during care provision. Improving access to care for transgender people is a multidimensional issue that should be addressed at the societal, healthcare, and research levels. Actions for future professional education initiatives should focus on respecting transgender identity, protecting confidentiality, creating trusted provider–patient relationships, and providing sufficient competency on trans-specific healthcare issues.

## 1. Introduction

According to the World Health Organization (WHO), the enjoyment of the highest attainable standard of health is one of the fundamental rights of every human being. However, trans people worldwide experience substantial health disparities and barriers to appropriate healthcare services that keep them from achieving the highest possible health status [1]. Barriers to healthcare experienced by trans communities include discriminatory treatment by healthcare providers, a lack of providers who are trained to offer appropriate healthcare, and refusal by many national health systems and health insurance programs to cover services for trans people [2]. Health disparities, as defined by differences in health outcomes among transgender individuals compared to cisgender individuals, have also been highlighted in the 2019 Behavioral Risk Factor Surveillance System report, which documents higher rates of adverse mental, physical, and behavioral health outcomes for transgender people. According to this report, 60% of transgender adults report having poor mental health at least one day in the past month compared with 37% of cisgender adults, while 54% of transgender adults report having had poor physical health at least one day in the past month compared with 36% of cisgender adults [3].

Access to healthcare is necessary to guarantee an adequate quality of life, not only to alleviate suffering, but also to ensure good health in the long run. Prejudicial attitudes among health professionals and inherent heteronormativity in health services can deter Lesbian, Gay, Bisexual, Trans, Queer, Intersex, and other persons (LGBTQI+) from seeking medical care. Additionally, protection of LGBTQI+ people against discrimination in healthcare and other services is limited in many European Union (EU) countries [4]. Although existing law implementing the principle of equal treatment between women and men is, to a certain extent, relevant to discrimination on grounds of gender identity, in many cases, there is no legal framework when it comes to discrimination on the basis of sexual orientation in any area outside employment [4,5,6,7,8].

Specifically for Greece, the problem is evident in all areas of life. Starting from the educational system, which is highly centralized and lacking sex education classes, attitudes towards LGBTQI+ people have been continuously negative, although this is slowly changing. Since 2009, the International LGBT Association (ILGA)-Europe has produced an annual benchmarking tool comprising the Rainbow Map and Index to illustrate the legal and policy situation of LGBTQI+ people in Europe. ILGA-Europe ranks 49 European countries by examining their laws and policies using a set of 69 criteria, divided between six thematic categories: equality and non-discrimination; family; hate crime and hate speech; legal gender recognition and bodily integrity; civil society space; and asylum. In the ILGA’s 2020 report, Greece was ranked 13th among the 49 European countries examined, with a 48% improvement in the Rainbow Map and Index compared to the previous year [9]. However, these law and policy improvements do not seem to reflect the general attitudes of the public. The 2019 report by the Pew Research Center indicated that 48% of Greek respondents believed that society should accept homosexuality, while a significant 47% did not believe so [10]. Younger people (18–29 years old), however, were found to be more tolerant than older people (50 years and older) regarding diversity.

Still, local particularities, including the strong and persistent societal influence of religion and church, the financial crisis that hit the country in the last decade, and the rise of far-right political views, have multiplied homophobic and transphobic incidents [9]. One of the most important measures against homophobia and transphobia that Greek LGBTQI+ organizations are bringing forward is education offered in schools, along with targeted training of public officials, including teachers, police officers, health care professionals, and clinic staff [11]. Furthermore, Greek LGBTQI+ organizations strongly advocate for a complete ban on practices that aim to change gender identity and/or sexual orientation (known as conversion therapies).

During the past four decades, trans identities have been classified as mental health disorders by both the International Classification of Diseases (ICD) and the Diagnostic and Statistical Manual of Mental Disorders (DSM). This pathologization has contributed immensely to the stigma trans people face when accessing health services, limiting their access as well as the rest of their rights (e.g., access to legal gender recognition). In 2013, the fifth edition of the *DSM* by the American Psychiatric Association (APA) renamed the category ‘Gender Identity Disorder’ to ‘Gender Dysphoria’, recognizing that trans identities are not per se pathological [12]. This edition also states that gender identity is a non-binary concept. The complete pathologization of trans identities occurred in May 2019, when the General Assembly of WHO upvoted the 11th edition of ICD, in which ‘gender identity disorders’ were substituted by ‘gender incongruence’ and all diagnoses related to trans identities were removed from the category ‘Mental and Behavioral Disorders’ and included in a new category named “categories related to sexual health” [13]. The aim was to remove the mental health stigma from trans identities while ensuring trans people’s access to gender-affirming procedures. Since 2022, the use of ICD-11 has been enforced in all countries, with 35 countries actually using it as of February 2022, according to the WHO [14].

In Greece, there is a generalized lack of research regarding the trans population, particularly their medical needs and access to health and social care services. This study is part of the European collaborative project TRANSCARE, which seeks to improve access to healthcare for transgender populations “https://transcare-project.eu/ (accessed on 10 January 2024)” through the implementation of a range of interrelated actions, including context mapping, primary field research, educational interventions, and policy recommendations. As part of TRANSCARE’s exploratory situation analyses and in the absence of particular and solid evidence from Greece, the present review aims to synthesize the international literature on barriers that transgender individuals face in terms of access to healthcare. Results are discussed on the basis of their relevance to Greece and on how they could be used to address training gaps and the needs of local healthcare professionals so as to ultimately contribute to improving the quality of care provided to the local transgender population.

## 2. Materials and Methods

### 2.1. Design

A scoping review was conducted to provide a synthesis of the evidence from diverse healthcare studies and help inform clinical practice and health policy [15]. Rather than evaluating or weighting the findings of individual studies, scoping reviews provide a snapshot of an overlooked or emergent field of research. The current scoping review was conducted following the five-stage methodological framework, which entails (a) identifying the research question; (b) identifying relevant studies; (c) selecting relevant studies; (d) charting the collected data; and (e) synthesizing, summarizing, and reporting the findings [16].

### 2.2. Search Strategy

The database of PubMed was used for the search of peer-reviewed articles and indexed publications (September–October 2020), since the focus of this issue was on the medical rather than psychological/social field (yet without excluding either somatic or mental healthcare articles). We also included gray literature in our search by hand-searching Google and Google Scholar. Gray literature was included in order to avoid missing reports and local documents, including Greek papers or European reports.

We searched in the English language for (synonyms of) the terms “transgender” (or “trans” or “transexual”) AND “barriers” OR “access” (or “accessibility) AND “healthcare” (or “healthcare delivery”). Details regarding the search strategy are provided in Appendix A. The term “transgender” (or “trans” or “transexual”) is used here as the umbrella term that describes the full range of people whose gender identity and/or gender role do not conform to what is typically associated with their sex assigned at birth.

All reports were inserted into the reference manager Mendeley to delete potential duplications. Two researchers assessed the relevance of the identified documents by screening the title, abstract, and subsequently the full text of the remaining papers. Articles were eligible for inclusion if they were:Reporting on at least one barrier to care for trans individuals or at least one educational need for healthcare providers;Had free full text availability;Published from 2015 and onwards, so as to retrieve recent evidence published within the last 10 years.

Documents were excluded if there was no full text available or if merely hypothetical factors were reported (i.e., if there was no actual investigation of barriers or healthcare/educational needs). With the exception of literature reviews, reports, doctoral dissertations, and other documents that did not include primary data (e.g., protocols, conference abstracts, etc.), they were excluded. The exceptions were applied due to the lack of relevant peer-reviewed information for the local context. If there was doubt about the inclusion/exclusion of a document, this was solved by discussion and consensus among the research team.

### 2.3. Appraisal of Methodological Quality

A subjective appraisal of the methodological quality of identified papers was performed using the international guidelines of the Enhancing the Quality and Transparency of Health Research (EUATOR) Network [17], which (depending on the situation) included (but were not limited to) the Strengthening the Reporting of Observational Studies (STROBE) statement [18], the Preferred Reporting Items for Systematic Reviews and Meta-Analyses (PRISMA) statement [19], and the Consolidated Criteria for Reporting Qualitative Research (COREQ) [20].

### 2.4. Data Extraction and Analysis

Of all the remaining documents, information on the authors, publication date, journal, type of study, and setting (location and population) was extracted. Outcomes, namely barriers and, subsequently, facilitators to care for transgender individuals, were extracted.

### 2.5. Reporting

This study is reported according to the PRISMA statement guidelines [19].

### 2.6. Ethics

As part of TRANSCARE, this study was approved by the Research Ethics Committee of the University of Crete (protocol number: 150/15.07.2020).

## 3. Results

### 3.1. General Characteristics

The initial PubMed search yielded 556 references and, additionally, four records from other sources. Of these 560 documents, 234 were screened for eligibility, while 124 full texts were assessed. Eventually, 69 were found relevant and included in the present analysis, with only three reporting empirical research from Greece. The study flowchart is presented in Figure 1.

General characteristics were extracted from the selected documents and are provided in the tables of Appendix A. The majority of identified research originated in the United States of America (USA) and featured qualitative methodology or survey data analysis. Very limited evidence was identified from Europe.

The wide range of information extracted from this literature review is summarized in Figure 2. Several individual-, interpersonal-, and institutional-level barriers were determined, while respective facilitators for addressing these barriers at the societal, health system, and research levels were also noted [21,22,23,24,25,26,27,28,29,30,31,32,33,34,35,36,37,38,39,40,41,42,43,44,45,46,47,48,49,50,51,52,53,54,55,56,57,58,59,60,61,62,63,64,65,66,67,68,69,70,71,72,73,74,75,76,77,78,79,80,81,82,83,84,85]. From Greece, documents were identified only in gray literature and mainly concerned general reports of LGBTQI+ organizations, with only one report [83] presenting research focused on understanding the health inequalities of LGBT people in the country.

### 3.2. Barriers at the Individual Level

One of the most frequently encountered barriers is related to the stigma that transgender people experience within healthcare services [2,23,27,30,31,34,44,45,46,51,52,61,69,73,80,84,85]. Stigma is related to discriminatory behaviors towards trans patients by healthcare providers, as well as feelings of anxiety and fear of negative encounters on behalf of trans patients. Both significantly hamper access to care, resulting in avoidance or delay in healthcare seeking on behalf of patients and suboptimal care or, even, denial of care provision on behalf of healthcare providers. Stigma and invisibility were also found to be the main factors that interrelated with participants’ experiences of low-quality health care services in the only study from Greece reporting empirical research on understanding the health inequalities of LGBTQI+ people in the country [83]. According to the same study, this invisibility is further contributing to patterns of homophobia, transphobia, and pathologization towards LGBTQI+ people, with the author concluding that “unless we overcome this invisibility and comprehend what it really means to be discriminated (…), the terms ‘sexual orientation’ and ‘gender identity’ as bases of discrimination will remain abstract in official documents regarding health rights” [83].

Studies also suggested that many transgender individuals may lack information about gender- and health-related issues, thus presenting low health risk perceptions, in the sense that certain high-risk behaviors may be perceived as less risky and followed at high rates without protective measures [2,44,46,51,52,67,83]. Additionally, they are frequently faced with limited awareness regarding the availability of trans-specific and other healthcare services [44,56,60,64]. These barriers may impede the search for competent care and contribute to worse health outcomes.

Social determinants of health, including poverty and lack of healthcare insurance, along with other factors such as older age and non-Caucasian race, were also identified as barriers that may postpone or prevent access to preventive care, medication, or treatment for many transgender individuals [26,28,42,44,46,52,55,61,68,74,76,77].

### 3.3. Barriers at the Interpersonal Level

The lack of knowledgeable and experienced healthcare providers was another significant factor impeding access to care, as reported in many studies [6,22,28,35,37,40,50,53,57,59]. The gap in professional training resulting from the lack of education on trans-inclusive healthcare, especially regarding sexual health services that trans people may need (e.g., access for trans men to pap smears or for trans women to prostate exams) and gender-affirmative issues (e.g., hormone therapy, surgery), along with a lack of exposure to the transgender population, has been highlighted by both transgender individuals and healthcare providers as an important area of focus. Lack of medical guidelines and protocols specific to transgender health also contributes to the limited competence of healthcare providers. One study further reports that transphobia among care providers, rather than education, predicts professional pursuit of knowledge on transgender health issues [57].

In the same direction, cultural competencies (e.g., using the appropriate names and pronouns) and communication skills, essential parts of professional education, were identified by many studies as necessary factors for enhancing healthcare seeking by transgender individuals [6,28,40,49,50,74,76,84].

Many of the studies suggested that transgender participants often face lack of sensitivity, erasure, or doubt of their trans identity by healthcare providers. They indicated that healthcare professionals often expressed negative attitudes and transphobic or discriminatory behaviors, such as referring to them in a gender other than the one they identify with (misgendering). In many cases, transgender participants reported being mistreated or even denied care [24,29,41,54,57,83]. Breaking confidentiality or privacy, gossiping, or focusing on retrieving unnecessary personal rather than clinical information were reported as forms of trans patients’ mistreatment [30,34,51,56,66,73].

In some studies, trans people reported that their gender identity is often perceived by healthcare professionals as a sign of pathology, a mental health disorder, a hormonal disorder, depression, or even a choice of cognitive control [30,41,53,66,83]. Such perceptions may lead to misdiagnoses and inappropriate treatments, psychological trauma, and avoidance of care.

In general, the lack of trusted, respectful, and non-judgmental environments in healthcare services was also identified as an important barrier by many studies [25,28,30,35,40,41,47,49,63,72,74,76]. Experiences or concerns of confidentiality loss and privacy violation account for substantial hesitations in disclosing trans identity and medical history during consultations, ultimately contributing to misdiagnoses, inappropriate care provision, and compromised health status for trans individuals.

### 3.4. Barriers at the Institutional Level

The persistent binary approach to gender and health prevailing in most healthcare systems also contributes to improper care delivery both in terms of general care provision and, particularly, gender-affirming procedures (e.g., hormonotherapy and surgeries), as it forces people to accept a gender diagnosis or conform to the gender assigned at birth without addressing trans-specific healthcare needs [23,37,39,46,47,48,51,61,63,68,75,76,80]. Still, even if trans identity disclosure is enabled, systems have not yet accounted for the mechanisms to appropriately collect and track information about transgender patients (e.g., non-binary medical history forms or registrations to electronic medical record systems).

Apart from the limited access to general healthcare services, the lack of services specifically for transgender individuals and their particular health needs (e.g., mental, sexual, reproductive, endocrinological, etc. health issues experienced specifically by transgender people) was noted in many studies as a substantial barrier to care for transgender individuals [29,35,39,50,51,60,64,81,83,84,85]. In the same context, many stretched the need for accounting for inclusive healthcare environments where transgender professionals are involved in care planning and decision making and where trans identity is visible and the general ambient aesthetic refrains from the stereotypical binary expression.

In the context of the high medical costs that may be related to the essential healthcare needs of transgender patients, appropriate coverage by healthcare insurance is identified as crucial [42,43,44,45,46,55,61,68,74,76,77]. In many cases, high-cost but necessary gender-affirming medical procedures or treatments (e.g., hormone therapies or surgeries) are not reimbursed by insurance companies, resulting in further exclusion and suboptimal outcomes for transgender individuals.

Fragmented systems also contribute to negative experiences with healthcare services [21,34,40,56,63,68,74]. In many studies, transgender individuals suggested that they would feel more empowered to seek care if they knew that their healthcare provider could properly refer them to other health and/or social care professionals or if they could effectively navigate through the healthcare system.

## 4. Discussion

### 4.1. Main Findings and Comparison with the Literature

Our scoping review identified several barriers to caring for transgender people. Our search yielded results that mainly originated from settings outside Europe; however, they are consistent with the limited evidence from the European setting. Similar to our findings, the 2020 results of the European Union Agency for Fundamental Rights (FRA) survey suggest that LGBTQI+ people may not disclose their identity due to fear of stigma or discrimination [86]. On average, 37% of those surveyed would not be open about their LGBTQI+ status with any healthcare personnel.

According to the same report, one in twelve (8%) of people who accessed social services felt personally discriminated against by healthcare personnel. Among transgender individuals, the level of discrimination was twice as high: almost one in five say they were discriminated against by healthcare personnel (19%). Moreover, respondents who were open to medical staff about being LGBTQI+ were at least 50% more likely than those who hide their LGBTQI+ identity to say they have experienced one of these situations [86]. Furthermore, in accordance with our results, a lack of culturally and clinically competent healthcare professionals is reflected in a 2020 nationally representative survey of LGBTQI+ adults in the US, where nearly half of transgender respondents experienced some form of mistreatment from a healthcare professional in the past year prior, including care refusal, misgendering, and verbal or physical abuse. In the same report, one in three transgender participants had to teach their doctor about transgender identity in order to receive appropriate care [85].

In line with our results, more recent evidence suggests that financial hardships disproportionally faced by transgender people. Results from the Center for American Progress (CAP) study highlight that 40% of transgender respondents reported postponing or avoiding preventive screenings in the past year due to cost, while more than half reported postponing or avoiding necessary medical care because they could not afford it [87]. Poverty, unemployment, and homelessness, still constitute a significant barrier to care for this population—a phenomenon that has been even more aggravated due to the recent COVID-19 pandemic [88,89]. Another recent large-scale US study comparing transgender (n = 1678) and cisgender adults (n = 403,414) from the 2019 to 2020 Behavioral Risk Factor Surveillance System found that transgender adults were significantly more likely to experience barriers to care in general and more likely to delay care due to costs [90].

In the same direction, insurance challenges still prevail among the barriers to care for transgender individuals, with both public and private insurance bodies denying coverage for essential gender-affirming care, increasing out-of-pocket costs for the population [91]. According to the CAP survey, 46% of transgender respondents reported having a health insurer deny them gender-affirming care, while 34% reported that a health insurance company refused to change their records to reflect their current name or gender [87]. Still, data from the first US national probability survey of transgender persons comparing both transgender and cisgender and 2021 versus 2015 data also suggest that health disparities and health access barriers are still prevalent among the trans population. Results highlight that transgender people more often avoid care due to cost concerns beyond their insurance status, while they more often rate their health as fair/poor, with more frequently occurring poor physical and mental health days compared to cisgender participants. Comparison of results with 2015 data showed no differences in health outcomes or access, reflecting the slow pace of change in improving care provision for transgender individuals [92].

Similar to our findings, health issues and needs that are of specific concern for transgender populations still remain unmet. Transgender people often pursue gender-affirming health care, such as hormone therapy and/or surgeries, facing significant challenges. In a 2023 systematic review of factors sociodemographic factors, treatment-related factors, psychosocial factors, and particularly health care interactions were identified as strong determinants of experiences transgender people report in regards to associated gender-affirming healthcare [93]. Another recent study linked gender-affirmative care to the regular issue faced by transgender individuals who, in many countries, must undergo a psychosocial assessment and receive a letter of support from a mental health care provider to access hormone and other specific treatment. In this study, transgender participants reported the increased psychological distress experienced during the psychosocial assessment as a significant barrier to gender-affirmative care [94].

Furthermore, transgender individuals face specific reproductive and sexual health needs that are also obstructed by significant barriers. In agreement with the barriers identified by our review, a recently published meta-synthesis of evidence on the fundamental problems that the transgender population faces regarding experiences of sexual and reproductive health illustrates the impact of individual, interpersonal, institutional, and social factors that influence this health aspect, including limited information, lack of awareness, low socioeconomic status, stigma, discrimination, and social deprivation [95]. Another recent literature review further emphasizes that research on specialized reproductive healthcare for transgender individuals (e.g., medical interventions and fertility, gamete preservation, etc.) is still not receiving proper attention, while routine interventions such as preconception counseling, prenatal surveillance, perinatal support, contraceptive, and pregnancy termination-related healthcare are still lacking meaningful adaptation for this patient population, and many knowledge gaps remain to be filled [96].

In the same direction, barriers to care in regards to sexually transmitted diseases, particularly HIV, pose specific challenges to the transgender population. In a large 2023 USA study focusing on indices of HIV risk among transgender women, unexpectedly high mortality was found. Although none of these deaths were HIV-related, the authors suggested that the issue is indicative of poor care and should not be neglected [97]. In terms of essential Pre-Exposure Prophylaxis (PrEP) medication for HIV prevention, another recent scoping review found high willingness (80%) to use PrEP among transgender women worldwide, yet low uptake and adherence (35.4%), with socioeconomic determinants, including poverty, substance use, stigma, and mistrust, again driving this phenomenon [98], in line with the barriers identified by this study.

Other studies have also examined barriers in mental health care for transgender individuals, another particularly challenging area for this population. One of the first systematic reviews to specifically explore obstacles to transgender mental health care identified major barriers that are similar to those also reported in this review and include personal concerns involving fear of being pathologized or stereotyped, objections to common therapeutic practices, and incompetent mental health professionals, including those who are unknowledgeable, unnuanced, and unsupportive, as well as affordability obstacles faced by transgender individuals [99].

In terms of common chronic diseases, such as cancer, a 2023 narrative scoping review focusing on transgender cancer care identified similar barriers to our results, suggesting suboptimal service delivery in the continuum of care. This review revealed that transgender people were not only burdened by high prevalences of risk factors (e.g., tobacco, alcohol) but also by high rates of HPV and HIV-related cancers. They were less likely to adhere to cancer screening, while socio-economic determinants seemed to drive these disparities. Lack of knowledgeable healthcare practitioners represented a major hurdle to cancer prevention, care, and survivorship for this population, while discrimination, discomfort caused by gender-labeled oncological services, stigma, and a lack of cultural sensitivity of health care practitioners were other barriers met in the oncology setting [100].

In line with our results, system-level barriers have also been documented in more recent research. In a qualitative study of US transgender adults, misgendering, system-wide insensitivity during health care encounters, and low levels of understanding of their transgender experience among primary care providers were mentioned as significant adversities. Provider–patient relationship improvements and service–provider sensitivity training. For better care coordination and enhancing patient support for the navigation of gender affirmation services, careful consideration when implementing systemwide routine processes such as using correct pronouns and names was recommended as ways to overcome these barriers [101].

Despite the limited evidence from Europe compared to the USA, results from a 2023 study evaluating experienced barriers to care for transgender individuals (n = 307) in three countries (Belgium, the Netherlands, and Germany) may be indicative of the European landscape. In this study, the majority of participants reported various barriers, with the most frequently reported being the lack of support from the social environment, travel time to clinics and costs, and stiff treatment protocols in the sense that many participants reported they had to convince their provider they needed care and/or express their wish in such a way to increase their likelihood of receiving care. More mental health problems, lower income, and female gender were again associated with more negative experiences, suggesting that European health systems need to address both personal and systemic characteristics to overcome barriers to transgender care [102].

### 4.2. Strengths and Limitations

To the best of our knowledge, this is among the first Greek studies to document barriers to care for transgender individuals and to discuss the development of interventions and strategies for enhancing the access of the transgender population to the country’s health system. However, our study has several limitations. Firstly, we have searched in only one database, potentially missing relevant papers that are not included in PubMed. Although we noticed the majority of our identified factors reoccurred throughout the papers (leading to data saturation), publications from other databases may potentially result in a shift in key themes. Additionally, although we based our search strategy on the terminology that was widely used during the time of our review, new guidance has been issued since then, particularly the Standards of Care (SOC) version 8, which introduced the term “Transgender and Gender Diverse People” instead. Finally, even if they may exist in other databases or they may be located via different search strategies, our particular search strategy did not manage to track many published, peer-reviewed papers from the European or Greek setting. However, this fact that we did not find many local studies might be as well attributed to the generalized lack of such particular evidence due to the very limited research about the trans population of Greece in general.

### 4.3. Implications for Professional Education in the Greek Context

Our results indicate that improving access to care for transgender people is a multidimensional issue that should be addressed at the societal, healthcare, and research levels. However, Greece faces substantial challenges in regards to meeting the above requirements. According to the Eurobarometer on Discrimination (2015) reports [103], Greece shows high rates of discrimination based on sexual orientation (71%) and gender identity (73%). The annual Greek Racist Violence Recording Network (RVRN) data from 2012 until 2017 shows high rates of verbal and/or physical violence, with 934 incidents of racist violence incidents concerning more than 1000 victims and, more specifically, 98 violent incidents against trans people between 2015 and 2018 [104,105,106]. In 2014, a law on hate crimes and hate speech was upvoted by the Greek parliament, prohibiting hate crimes and hate speech on the basis of gender identity (N. 4285/2014). Two years later, the legal framework on equal treatment and combating discrimination (N. 4443/2016) was also updated to include a prohibition of all forms of discrimination on the basis of gender identity, but only in the sectors of employment and accessing services and goods [107].

According to Giannou (2017), trans people also seem to be excluded from healthcare due to negative stereotypes, stigma, and general social assumptions [83]. One of the most important steps regarding the protection of trans people’s rights in Greece was the introduction of a new law on legal gender recognition in 2017 (L. 4491/2017), which allowed trans people to change their name and gender on their official documents without the requirement for psychiatric evaluation or other medical procedures (e.g., hormone therapy or gender-affirming surgeries). However, the law has several shortcomings, among which the costly and lengthy procedure trans people are required to follow, the lack of gender options outside the binary, and the exclusion of people who are married, as well as underage persons under 17 years old. As a result, despite the current legal reform, not all trans people can (easily) access official documents that reflect their gender identity [108].

As noted in the results, for trans people who wish to pursue gender-affirming medical interventions such as hormone-replacement therapy or surgeries, the high cost and general lack of specialized health professionals can be important obstacles. In Greece, hormone-replacement therapy is covered by social health insurance, whereas all kinds of gender-affirming surgery are not. It is important to note that until ICD-11 is enforced in Greece, trans people who want to access medical transition procedures are required to receive a psychiatric diagnosis of “Gender dysphoria” [108].

Furthermore, problems and contextual issues that have been observed in the implementation of healthcare services in Greece, including the lack of integrated care and patient-centered approaches, could affect the establishment of effective doctor–patient relationships and, therefore, the delivery of appropriate care for transgender people by disregarding their healthcare needs [83,109]. Primary care has been proven to provide a fruitful ground for designing and implementing novel approaches to enhance care for transgender individuals [110].

Finally, there is no academic training focused on transgender issues in medical or health professional programs, sustaining the knowledge gap among healthcare professionals. However, specific training hours could be dedicated to existing courses that address doctor–patient communication or the development of interpersonal skills. The further lack of Greek guidelines on transgender health also contributes to the limited promotion of good health, best practices, and healthcare outcomes for this population. Future professional training initiatives will benefit from the involvement of transgender people in design and delivery and should focus on:Respecting transgender identity in clinical environments;Protecting confidentiality and creating trusted provider–patient relationships under effective communication styles;Providing sufficient knowledge (theory or practice) on trans-specific healthcare issues and needs;Advocating for patients, especially with other providers/services.

The evidence on the development and implementation of interventions to tackle barriers to care for transgender individuals has grown in the last few years. In particular, a literature review focusing on HIV care for transgender individuals identified promising interventions that address structural and individual barriers, including societal and cultural stigma. The content of the strategies presented in this review varied from providing HIV testing and linkage to HIV care and social services through peer navigators to establishing group sessions focusing on emotional and cognitive processing of trauma, coping, and resilience for transgender women and to providing housing, employment, and legal services to this population. The review highlights the fact that all these interventions are built on the basis of ensuring the meaningful participation of the trans community in their design and implementation, the development of programs that ensure the integration of gender-affirming care and social services with HIV care, the focus on improving behavioral health outcomes, the deployment of peer-led counseling, education, and navigation, and the provision of technology-based interventions to increase access to care [111].

Another extensive literature review provides detailed information regarding the research on educational interventions conducted so far, with the aim of reducing barriers to care for transgender individuals [112]. The content of the reported educational interventions is focused on transgender and LGBTQI+ health and is addressed to medical students and/or practicing healthcare professionals. The delivery formats of these interventions vary from workshops, elective courses, curricular additions, and simulation activities for students to face-to-face or online continuous medical education programs for experienced professionals, with the review highlighting the benefit of all these actions on raising professionals’ knowledge, skills, and quality of provided services [112].

As part of the TRANSCARE project, the results of this study will be used to adapt international best practices and to further inform local activities, including large-scale needs assessment surveys to collect primary data on the educational needs of Greek medical students and healthcare professionals, the ultimate development of the first evidence-based continuous medical education program for transgender health in Greece, the delivery of nation-wide information and awareness-raising activities, and the release of full-scale policy recommendation reports. The integration of educational programs and health policy recommendations into the local healthcare system is crucial towards achieving community-integrated health, and there are several frameworks and methods proposed by leading organizations such as the WHO [113] that are being used to guide this process throughout TRANSCARE. Taking into consideration the extensive primary healthcare reform that could provide the ground for hosting such interventions, along with the changes in the curriculum of general practice currently unfolding in the country [114], this study comes as timely as ever to provide space for action on transgender health in Greece.

## 5. Conclusions

Improving access to care for transgender people is a multidimensional issue that should tackle significant barriers at the personal, interpersonal, and institutional levels. Our review suggested that actions for future professional education initiatives should focus, among others, on respecting transgender identity, protecting confidentiality, creating trusted provider–patient relationships, and providing sufficient competency on trans-specific healthcare issues.

## Figures and Tables

**Figure 1 healthcare-12-00647-f001:**
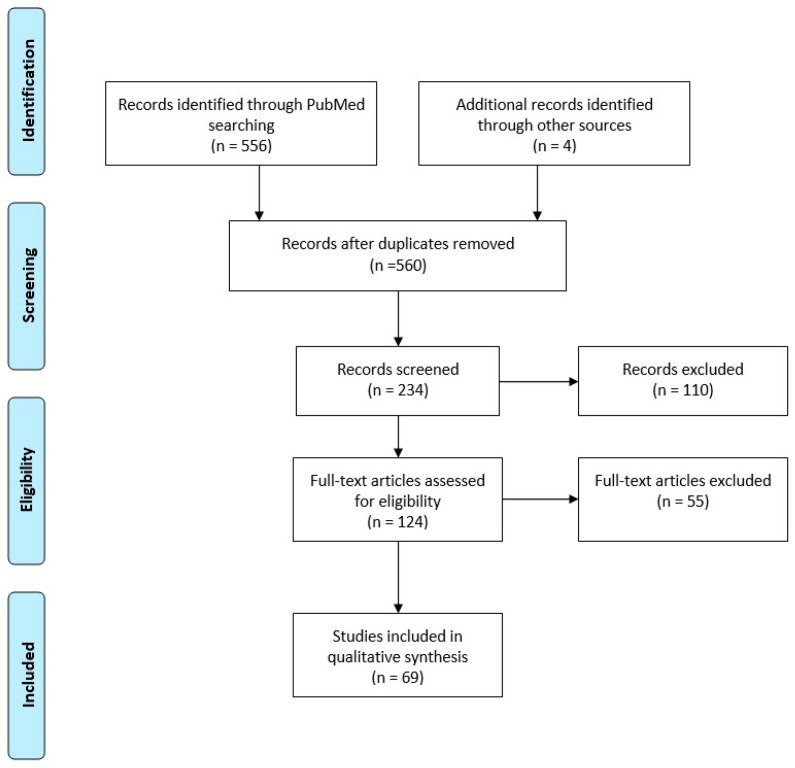
Flow diagram of document selection.

**Figure 2 healthcare-12-00647-f002:**
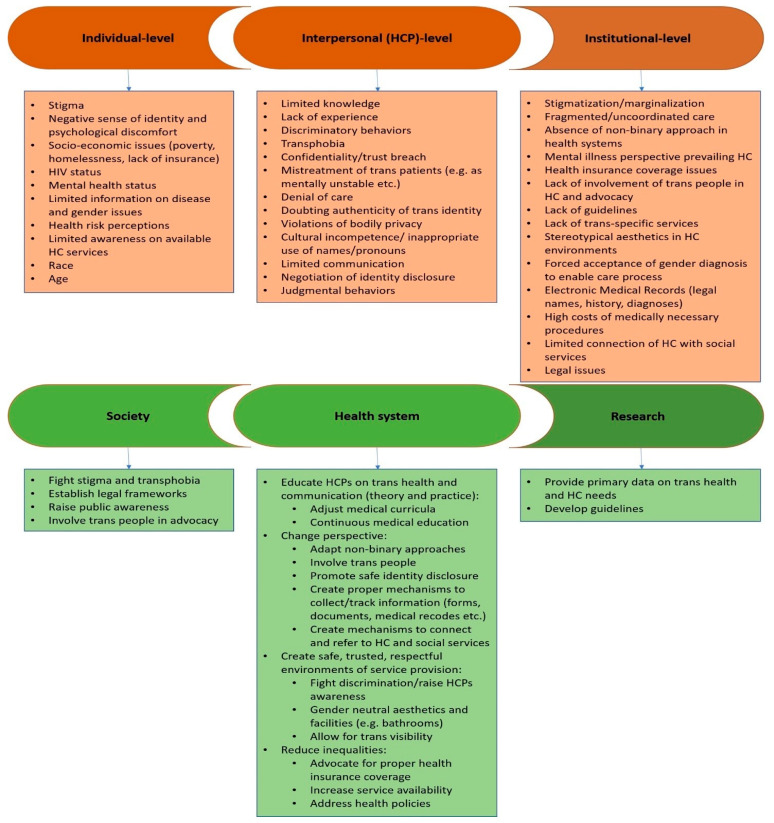
Barriers (red) and facilitators (green) to care for transgender individuals, as identified in this review. Abbreviations: HCP: healthcare provider; HC: healthcare.

## Data Availability

Not applicable.

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
