# Peer review of "Βarriers and Gaps to Medical Care for Transgender Individuals: A TRANSCARE Scoping Review with a Focus on Greece"

_healthcare, 2024, doi:10.3390/healthcare12060647_

Round 1
Reviewer 1 Report (Previous Reviewer 2)
Comments and Suggestions for Authors
Thanks to the authors for taken the comments of this reviewer seriously. In general, this manuscript improved considerably. Only some minor problems remain:
1.Line 19-20: “Out of 560 identified references, 69 were included in this study.” It would be informative for the reader, given the focus on Greece, to insert here how many of these papers report empirical research on Greece. For example: Of these 69 publications only N report empirical research on Greece.
2. Line 60-68: “In a 2020 report from the International LGBT Association (ILGA) which assessed LGBTQI+ rights Europe, Greece was ranked 13th among 49 other European countries, with 48% improvement regarding legal and policy practices towards trans individuals in the thematic categories of quality and non-discrimination, family, hate crime and hate speech, legal gender recognition and bodily integrity, civil society space and asylum (9). However, this does not reflect the change in general attitudes of the public. The 2019 report by the Pew Research Center indicated that 48% of Greek respondents believed that society should accept homosexuality, while 47% did not believe so (10). Younger people (18-29 years old) were found more tolerant than older people (50 years and older) regarding diversity”.
I do not understand “with 48% improvement regarding legal and policy practices”. What was the measurement-unit? Neither do I understand why “However, this does not reflect the change in general attitudes of the public. The 2019 report by the Pew Research Center indicated that” follows logically from “with 48% improvement regarding legal and policy practices”. Please clarify.
3.Line 114-116: “The database of PubMed was used for the search of peer-reviewed articles and indexed publications (September – October 2020), since the focus of this issue was on the medical, rather than phycological/social field (yet without excluding either somatic or mental healthcare articles).” Given the title of the paper, “Βarriers and gaps to care for transgender individuals: a TRANSCARE scoping review with a focus on Greece”, and especially the use of the general category “care”, I find from a methodological point of view a search strategy that only use PubMed a reductionistic and weak approach (as also discussed by the authors in the discussion of this paper). So, I suggest that the authors change the title in: Βarriers and gaps to medical care for transgender individuals: a TRANSCARE scoping review with a focus on Greece”.
And “phycological/social field” should be: psychological/social field
4. Line 172-174: “From Greece, documents were identified only in gray literature and mainly concerned general reports of LGBTQI+ organizations with only one report presenting research focused on understanding the health inequalities of LGBT people in the country.” Insert specific reference to this one report on empirical research in Greece. And describe the specific results of this specific study in the Results section.
5. Line 291: If you use CAP here for the first time, please do not use the abbreviation.
6. Line 387: “temic characteristics to overcome barriers to transgender care (102).4.2. Strengths and lim- itations” Strengths and limitations should begin at a new line.
7.Line 393-395: “Although we noticed the majority of our identified factors reoccurred throughout the papers (leading to data saturation), publications from other databases may potentially result in a slight shift in key themes”. Given that other databases were not consulted, you do not know in advance if an eventual shift would be small or big. So, delete slight.
8.Line 396-398: “Our search strategy did not also manage to track many published, peer-reviewed papers from the European or Greek setting, although this might be attributed to the generalized lack of such particular evidence.” I do not understand this sentence: if there are no studies, they cannot be included in a review! And in consequence, they are not missed. Please clarify, or make a correction.
9.Line 456-475: “The evidence on the development and implementation of interventions to tackle barriers to care for transgender individuals is growing in the last years, while ongoing studies will determine the efficacy and effectiveness of these interventions . In particular, a literature review focusing on HIV care for transgender individuals identified promising interventions that address structural and individual barriers including societal and cultural stigma and highlight the fact that such interventions are built on the basis of ensuring the meaningful participation of the trans community in their design and implementation, the development of programs that ensure the integration gender-affirming care and social services with HIV care, the focus on improving behavioral health outcomes, the deployment of peer-led counseling, education, and navigation and the provision of technology-based interventions to increase access to care (111). Another extensive literature review provides detailed information on intervention research conducted so far with the aim to reduce barriers to care for transgender individuals, highlighting the significant potential of interventions that focus raising healthcare professionals’ education and skills, including educational interventions in medical students and residents and continuous medical education workshops, courses, curricular additions, simulation activities and other exercises (112). Taking into consideration the extensive Primary Healthcare Reform which could provide the ground for hosting such interventions, along with the changes of the curriculum of General Practice currently unfolding in the country (113), this study comes as timely as ever to provide space for action on transgender health in Greece.”
Does the first sentence imply that that now used interventions are NOT supported by empirical research-results? If that is the case, write it clearly! Furthermore, insert after ongoing studies the references of these studies. Mention also the content of the promising interventions, and not only that there are promising interventions.
10.Line 123-125: “The term “transgender” (or “trans” or “transexual”) is used here as the umbrella term that describes the full range of people whose gender identity and/or gender role do not conform to what is typically associated with their sex assigned at birth.” I find it remarkably the authors do not use the SOC-8 terminology “transgender and gender diverse individuals” . An explanation for the use of an idiosyncratic alternative terminology is not given. Although I would prefer such an explanation in the manuscript, the authors use of their own (defined) terminology is justified.
Author Response
Reviewer 1:
Thanks to the authors for taken the comments of this reviewer seriously. In general, this manuscript improved considerably. Only some minor problems remain:
Response: We thank the reviewer and we hope that our responses to the current round of feedback also achieve to improve our manuscript.
1.Line 19-20: “Out of 560 identified references, 69 were included in this study.” It would be informative for the reader, given the focus on Greece, to insert here how many of these papers report empirical research on Greece. For example: Of these 69 publications only N report empirical research on Greece.
Response: Thank you and we have added that only three studies report empirical research from Greece in both the abstract and the results section.
- Line 60-68: “In a 2020 report from the International LGBT Association (ILGA) which assessed LGBTQI+ rights Europe, Greece was ranked 13th among 49 other European countries, with 48% improvement regarding legal and policy practices towards trans individuals in the thematic categories of quality and non-discrimination, family, hate crime and hate speech, legal gender recognition and bodily integrity, civil society space and asylum (9). However, this does not reflect the change in general attitudes of the public. The 2019 report by the Pew Research Center indicated that 48% of Greek respondents believed that society should accept homosexuality, while 47% did not believe so (10). Younger people (18-29 years old) were found more tolerant than older people (50 years and older) regarding diversity”. I do not understand “with 48% improvement regarding legal and policy practices”. What was the measurement-unit? Neither do I understand why “However, this does not reflect the change in general attitudes of the public. The 2019 report by the Pew Research Center indicated that” follows logically from “with 48% improvement regarding legal and policy practices”. Please clarify.
Response: We thank the reviewer and we have re-written this section, hoping to have now clarified its meaning. The paragraph now reads: “Since 2009 the International LGBT Association (ILGA)-Europe has produced an annual benchmarking tool comprising of the Rainbow Map and Index to illustrate the legal and policy situation of LGBTQI+ people in Europe. ILGA-Europe ranks 49 European countries by examining their laws and policies using a set of 69 criteria, divided between six thematic categories: equality and non-discrimination; family; hate crime and hate speech; legal gender recognition and bodily integrity; civil society space; and asylum. In ILGA’s 2020 report, Greece was ranked 13th among the 49 European countries examined, with 48% improvement in the Rainbow Map and Index compared to the previous year (9). However, these law and policy improvements do not seem to reflect the general attitudes of the public. The 2019 report by the Pew Research Center indicated that 48% of Greek respondents believed that society should accept homosexuality, while a significant 47% did not believe so (10). Younger people (18-29 years old), however, were found more tolerant than older people (50 years and older) regarding diversity”.
3.Line 114-116: “The database of PubMed was used for the search of peer-reviewed articles and indexed publications (September – October 2020), since the focus of this issue was on the medical, rather than phycological/social field (yet without excluding either somatic or mental healthcare articles).” Given the title of the paper, “Βarriers and gaps to care for transgender individuals: a TRANSCARE scoping review with a focus on Greece”, and especially the use of the general category “care”, I find from a methodological point of view a search strategy that only use PubMed a reductionistic and weak approach (as also discussed by the authors in the discussion of this paper). So, I suggest that the authors change the title in: Βarriers and gaps to medical care for transgender individuals: a TRANSCARE scoping review with a focus on Greece”. And “phycological/social field” should be: psychological/social field.
Response: We thank the reviewer for this comment and we have acted as suggested by adding the word “medical” in the title and correcting the word “psychological”.
- Line 172-174: “From Greece, documents were identified only in gray literature and mainly concerned general reports of LGBTQI+ organizations with only one report presenting research focused on understanding the health inequalities of LGBT people in the country.” Insert specific reference to this one report on empirical research in Greece. And describe the specific results of this specific study in the Results section.
Response: Thank you and we inserted the specific reference as well as added the following text to the results section: “Stigma and invisibility were also found as the main factors that interrelated with participants’ experience of low-quality health care services in the only study from Greece reporting empirical research on understanding the health inequalities of LGBTQI+ people in the country (83). According to the same study, this invisibility is further contributing to patterns of homophobia, transphobia and pathologization towards LGBTQI+ people, with the author concluding that “unless we overcome this invisibility and comprehend what it really means to be discriminated (…), the terms ‘sexual orientation’ and ‘gender identity’ as bases of discrimination will remain abstract in official documents regarding health rights” (83)”.
- Line 291: If you use CAP here for the first time, please do not use the abbreviation.
Response: Thank you and we have now explained the abbreviation CAP (Center for American Progress) at first use.
- Line 387: “temic characteristics to overcome barriers to transgender care (102).4.2. Strengths and lim-itations” Strengths and limitations should begin at a new line.
Response: We thank the reviewer for noticing this and we have fixed the change in the respective line.
7.Line 393-395: “Although we noticed the majority of our identified factors reoccurred throughout the papers (leading to data saturation), publications from other databases may potentially result in a slight shift in key themes”. Given that other databases were not consulted, you do not know in advance if an eventual shift would be small or big. So, delete slight.
Response: Thank you and we deleted the word “slight”.
8.Line 396-398: “Our search strategy did not also manage to track many published, peer-reviewed papers from the European or Greek setting, although this might be attributed to the generalized lack of such particular evidence.” I do not understand this sentence: if there are no studies, they cannot be included in a review! And in consequence, they are not missed. Please clarify, or make a correction.
Response: Thank you and we rephrased this limitation to clarify its meaning. The sentence now reads: “Finaly, even if they may exist in other databases or they may be located via different search strategies, our particular search strategy did not manage to track many published, peer-reviewed papers from the European or Greek setting. However, this fact that we did not find many local studies might be as well attributed to the generalized lack of such particular evidence due to the very limited research about the trans population of Greece in general”.
9.Line 456-475: “The evidence on the development and implementation of interventions to tackle barriers to care for transgender individuals is growing in the last years, while ongoing studies will determine the efficacy and effectiveness of these interventions . In particular, a literature review focusing on HIV care for transgender individuals identified promising interventions that address structural and individual barriers including societal and cultural stigma and highlight the fact that such interventions are built on the basis of ensuring the meaningful participation of the trans community in their design and implementation, the development of programs that ensure the integration gender-affirming care and social services with HIV care, the focus on improving behavioral health outcomes, the deployment of peer-led counseling, education, and navigation and the provision of technology-based interventions to increase access to care (111). Another extensive literature review provides detailed information on intervention research conducted so far with the aim to reduce barriers to care for transgender individuals, highlighting the significant potential of interventions that focus raising healthcare professionals’ education and skills, including educational interventions in medical students and residents and continuous medical education workshops, courses, curricular additions, simulation activities and other exercises (112). Taking into consideration the extensive Primary Healthcare Reform which could provide the ground for hosting such interventions, along with the changes of the curriculum of General Practice currently unfolding in the country (113), this study comes as timely as ever to provide space for action on transgender health in Greece.” Does the first sentence imply that that now used interventions are NOT supported by empirical research-results? If that is the case, write it clearly! Furthermore, insert after ongoing studies the references of these studies. Mention also the content of the promising interventions, and not only that there are promising interventions.
Response: Thank you. To avoid confusion, we deleted the second half of the first sentence, as the point was not that now used interventions are not supported by empirical research results but rather that they are new and usually conducted in small samples - thus needing larger implementation and evaluation to be deemed effective. In terms of references, we cite the literature review papers (refs. 111 and 112) where the individual references of the reported interventions can be found, since our own reference list is already too long to add so many new papers. In terms of content, we added information on the content of the respective interventions. The section now reads: “The evidence on the development and implementation of interventions to tackle barriers to care for transgender individuals is growing in the last years. In particular, a literature review focusing on HIV care for transgender individuals identified promising interventions that address structural and individual barriers including societal and cultural stigma. The content of the strategies presented in this review varied from providing HIV testing and linkage to HIV care and social services through peer navigators to establishing group sessions focusing on emotional and cognitive processing of trauma, coping, and resilience for transgender women and to providing housing, employment and legal services to this population. The review highlights the fact that all these interventions are built on the basis of ensuring the meaningful participation of the trans community in their design and implementation, the development of programs that ensure the integration gender-affirming care and social services with HIV care, the focus on improving behavioral health outcomes, the deployment of peer-led counseling, education, and navigation and the provision of technology-based interventions to increase access to care (111). Another extensive literature review provides detailed information regarding the research on educational interventions conducted so far, with the aim to reduce barriers to care for transgender individuals (112). The content of the reported educational interventions is focused on transgender and LGBTQI+ health and is addressed to medical students and/or practicing healthcare professionals. The delivery formats of these interventions vary from workshops, elective courses, curricular additions and simulation activities for students to face-to-face or online continuous medical education programs for experienced professionals, with the review highlighting the benefit of all these actions on raising professionals’ knowledge, skills and quality of provided services (112). Taking into consideration the extensive Primary Healthcare Reform which could provide the ground for hosting such interventions, along with the changes of the curriculum of General Practice currently unfolding in the country (113), this study comes as timely as ever to provide space for action on transgender health in Greece”.
10.Line 123-125: “The term “transgender” (or “trans” or “transexual”) is used here as the umbrella term that describes the full range of people whose gender identity and/or gender role do not conform to what is typically associated with their sex assigned at birth.” I find it remarkably the authors do not use the SOC-8 terminology “transgender and gender diverse individuals”. An explanation for the use of an idiosyncratic alternative terminology is not given. Although I would prefer such an explanation in the manuscript, the authors use of their own (defined) terminology is justified.
Response: We thank the reviewer for this important comment. SOC-8 was released two years after the conduction of our review. We based our search on the terminology that was widely used during the time of our research. We however acknowledge the significance of the reviewer’s comment and have thus added a respective sentence in the limitations section reading: “Additionally, although we based our search strategy on the terminology that was widely used during the time of our research, new guidance has been issued since then and particularly the Standards of Care (SOC) version 8 which introduced the term ”Transgender and Gender Diverse People” instead”.
Reviewer 2 Report (New Reviewer)
Comments and Suggestions for Authors
I was glad to review tha paper titled "Barriers and Gaps to Care for Transgender Individuals: A TRANSCARE Scoping Review with a Focus on Greece". It presents a comprehensive scoping review on the barriers and gaps to care for transgender individuals, with a particular focus on Greece. Overall, the paper is clear and the methodology employed is appropriate. However, there are several aspects that require significant revision.
The primary critic aspect concern revolves around the lack of clarity regarding the connection between the international literature review and the policy recommendations for Greece. While each study included in the review provides insights into the health status of transgender individuals, considering the specificities of various regions, socio-health systems, and regulations, the relevance of adapting these findings solely to the Greek context remains unclear.
In this sense, the paper would benefit from a clearer articulation of its objectives. Specifically, it should clarify the rationale behind conducting a broad international literature review if the ultimate aim is to derive policy recommendations solely for Greece.
On this point, while the paper mentions policy implications, it would be beneficial to elaborate further on how the findings from the scoping review can inform policy development in Greece specifically. This could involve discussing potential adaptations or modifications of international best practices to suit the Greek context.
In addition, although the paper primarily addresses the transgender population, the introductory section initially encompasses a broader discussion on the LGBTQ+ community as a whole. In a specific point, the authors briefly touch upon levels of societal acceptance regarding homosexuality within the general Greek population. This approach risks confusing readers who may blur the distinction between gender identity and sexual orientation. It's imperative to revise this section, ensuring it exclusively focuses on issues pertinent to transgender individuals.
Finally, Several interesting aspects emerged from the analysis. However, in the discussion section, these points are not adequately linked to the literature under review. The authors frequently use vague expressions, such as "Many of the studies suggested...". It's essential to specify which studies are being referenced, including details such as the context in which they were conducted and the publication year. Each statement should be enriched with contextual information, and references to the specific texts emphasizing each aspect should be provided in parentheses.
Author Response
Reviewer 2:
I was glad to review the paper titled "Barriers and Gaps to Care for Transgender Individuals: A TRANSCARE Scoping Review with a Focus on Greece". It presents a comprehensive scoping review on the barriers and gaps to care for transgender individuals, with a particular focus on Greece. Overall, the paper is clear and the methodology employed is appropriate. However, there are several aspects that require significant revision.
Response: We thank the reviewer for the feedback and we hope our revisions to have improved our manuscript.
The primary critic aspect concern revolves around the lack of clarity regarding the connection between the international literature review and the policy recommendations for Greece. While each study included in the review provides insights into the health status of transgender individuals, considering the specificities of various regions, socio-health systems, and regulations, the relevance of adapting these findings solely to the Greek context remains unclear. In this sense, the paper would benefit from a clearer articulation of its objectives. Specifically, it should clarify the rationale behind conducting a broad international literature review if the ultimate aim is to derive policy recommendations solely for Greece.
Response: Many thanks for this important comment. We have elaborated on our aim by re-writing the last paragraph of the Introduction section to clarify that this study was just an initial step of a larger project that included subsequent primary field research and, mainly, that an international review was the only option due to the generalized lack of research regarding the Greek trans population (only three reports were identified with empirical data form Greece, as we also added in both the Abstract and Results’ section). The paragraph now reads: “In Greece, there is a generalized lack of research regarding the trans population and particularly their medical needs and access to health and social care services This study is part of the European collaborative project TRANSCARE which seeks to improve access to healthcare for transgender populations (https://transcare-project.eu/), through implementing a range of interrelated actions including context mapping, primary field research, educational interventions and policy recommendations. As part of TRANSCARE’s exploratory situation analyses and in the absence of particular and solid evidence from Greece, the present review aims to synthesize the international literature on barriers that transgender individuals face in terms of access to healthcare. Results are discussed on the basis of their relevance to Greece and on how they could be used to address training gaps and needs of local healthcare professionals, so as to ultimately contribute to improving the quality of care provided to the local transgender population”.
On this point, while the paper mentions policy implications, it would be beneficial to elaborate further on how the findings from the scoping review can inform policy development in Greece specifically. This could involve discussing potential adaptations or modifications of international best practices to suit the Greek context.
Response: Thank you. We have added text in the last paragraph of the discussion section and a new reference to highlight how the results of this review are being used locally. The paragraph now reads “As part of the TRANSCARE project, the results of this study will be used to adapt international best-practices and to further inform local activities including large-scale needs assessment surveys to collect primary data on educational needs of Greek medical students and healthcare professionals, the ultimate development of the first evidence-based continuous medical education program for transgender health in Greece, the delivery of nation-wide information and awareness raising activities and the release of full-scale policy recommendation reports. The integration of educational programs and health policy recommendations into the local healthcare system is crucial towards achieving community-integrated health and there are several frameworks and methods proposed by leading organizations such as the WHO (113) that are being used to guide this process throughout TRANSCARE. Taking into consideration the extensive Primary Healthcare Reform which could provide the ground for hosting such interventions, along with the changes of the curriculum of General Practice currently unfolding in the country (114), this study comes as timely as ever to provide space for action on transgender health in Greece”.
In addition, although the paper primarily addresses the transgender population, the introductory section initially encompasses a broader discussion on the LGBTQ+ community as a whole. In a specific point, the authors briefly touch upon levels of societal acceptance regarding homosexuality within the general Greek population. This approach risks confusing readers who may blur the distinction between gender identity and sexual orientation. It's imperative to revise this section, ensuring it exclusively focuses on issues pertinent to transgender individuals.
Response: We thank the reviewer. Our Introduction section includes information on the broader LGBTQI+ community because research on the local trans population exclusively is extremely limited. As we added in both our Abstract and Results’ section, it is indicative that this review identified only three reports Greece and all of these contain boarder data on LGBTQI+ in addition to data on trans individuals. As also per reviewer 1’s request, we performed changes in the Introduction section to clarify the data presented and, as mentioned above, we elaborated on the aim of our study to highlight that there is limited evidence from Greece that does not allow us to have a particular view on the Greek transgender population solely. We retain the rest of the information of the introduction section as we believe that it provides an illustrative presentation of the general situation in Greece and we hope that our rationale is now better delivered.
Finally, several interesting aspects emerged from the analysis. However, in the discussion section, these points are not adequately linked to the literature under review. The authors frequently use vague expressions, such as "Many of the studies suggested...". It's essential to specify which studies are being referenced, including details such as the context in which they were conducted and the publication year. Each statement should be enriched with contextual information, and references to the specific texts emphasizing each aspect should be provided in parentheses.
Response: We thank the reviewer. Since in the discussion section we attempt a comparison of our results with other studies rather than a repetition of identified data, we act as per reviewer’s request in the results section. In particular, we added specific references for the studies mentioned and, where most relevant, we included more contextual information, given the journal’s word count limitations. Detailed information on the publication year, study type, population, country/setting, area of focus and reported outcomes is provided for all 69 identified studies in Supplementary Table 2.
Round 2
Reviewer 2 Report (New Reviewer)
Comments and Suggestions for Authors
I am glad to see that the authors have improved the paper, considering most of the suggestions I had made. So, in the present form, I believe the paper is suitable for publication.
This manuscript is a resubmission of an earlier submission. The following is a list of the peer review reports and author responses from that submission.
Round 1
Reviewer 1 Report
Comments and Suggestions for Authors
The paper is an important piece of work contributing to the trans health studies. It also fills the knowledge gap on service accessibilities of trans people. I have the following suggestions, for enhancing the paper publication:
1) The most updated reference cited is 2020, will there be more publications (2020-2023) related to trans health care that can incorporate into the discussion section, to link with the results and previous literature at the introduction part?
2) The results and title of this article are related to transgender health, and the discussion section is not focusing on transgender health (broader issue on LGBT health), it appears to give the readers an impression of inconsistencies. Suggest relating the trans health publications in recent 3-4 years to your discussion of results. Trans and LGBQ service users may have overlapping concerns, but trans people, especially when going to health care needs, have significantly pressing and diverse medical needs. The authors may need to take care of this concern when presenting their discussions and implications with the results.
3) The focus of the paper (as observed from the title) is on Greece, will the authors consider giving a brief highlight on the cultural and political/social barriers for trans (if possible) or LGBT people to access health care in Greece specifically? in the introduction and the discussion sections?
Thank you for giving me the opportunity to read your paper.
Author Response
Reviewer 1
The paper is an important piece of work contributing to the trans health studies. It also fills the knowledge gap on service accessibilities of trans people. I have the following suggestions, for enhancing the paper publication:
Response: We thank the reviewer for the important commentary which has substantially helped us improve our manuscript. We hope to have now addressed all considerations.
1) The most updated reference cited is 2020, will there be more publications (2020-2023) related to trans health care that can incorporate into the discussion section, to link with the results and previous literature at the introduction part?
Response: We thank the reviewer and we also acknowledge the need to update our references, since a rapidly increasing amount of evidence can be found in literature from 2021 onwards. We particulary included the results of 16 newly added references (published since 2021) in our Discussion section to compare and link them with our own findings. Since we feel that this comment is directly related to your immediately following one, please, find the detailed information regarding the new text that we inserted in our manuscript in our response to your next comment (comment with number 2).
2) The results and title of this article are related to transgender health, and the discussion section is not focusing on transgender health (broader issue on LGBT health), it appears to give the readers an impression of inconsistencies. Suggest relating the trans health publications in recent 3-4 years to your discussion of results. Trans and LGBQ service users may have overlapping concerns, but trans people, especially when going to health care needs, have significantly pressing and diverse medical needs. The authors may need to take care of this concern when presenting their discussions and implications with the results.
Response: We thank the reviewer and, as per our response in comment 1, we have added a significant amount of new evidence to address this issue in our Disucssion section. Namely, we compared our results with more recent studies reporting on barriers to care for transgender individuals both in general and around several pressing and diverse trans-specific healthcare needs including gender-affirming processes, reproductive and sexual health (including HIV), mental health, cancer and other barriers related to systemic factors. As such, the following text has been added in our Discussion section: “Furthermore, in accordance to our results, lack of culturally and clinically competent healthcare professionals is reflected in a 2020 nationally representative survey of LGBTQI+ adults in the US where nearly half transgender respondents experiencing some form mistreatment from a healthcare professional in the past year prior, including care re-fusal, misgendering, and verbal or physical abuse. In the same report, one in 3 transgender participants had to teach their doctor about transgender identity in order to receive appropriate care (85).
In line with our results, more recent evidence suggests that financial hardships dispropor-tionally faced by transgender people, including poverty, unemployment and homeless-ness still constitute a significant barrier to care for this population - a phenomenon that has been even more aggravated due to the recent COVID-19 pandemic (86, 87). Results from the CAP study highlight that 40% of transgender respondents reported postponing or avoiding preventive screenings in the past year due to cost, while more than half report postponing or avoiding necessary medical care because they could not afford it (85). An-other recent large-scale US study comparing transgender (n = 1,678) and cisgender adults (n = 403,414) from the 2019 to 2020 Behavioral Risk Factor Surveillance System found that transgender adults were significantly more likely to experience barriers to care in general and more likely to delay care due to costs (88).
To the same direction, insurance challenges still prevail among the barriers to care for transgender individuals., with both public and private insurance bodies denying coverage for essential gender affirming care, increasing out-of-pocket costs for the population (89). According to the CAP survey, 46% of transgender respondents reported having a health insurer deny them gender-affirming care, while 34% reported that a health insurance company refused to change their records to reflect their current name or gender (85). Still, data from the first US national probability survey of transgender persons comparing both transgender to cisgender and 2021 versus 2015 data also suggest that health disparities and health access barriers are still prevalent among the trans population. Results high-light that transgender people more often avoid care due to cost concerns, beyond insur-ance status, while they more often rate their health as fair/poor, with more frequently oc-curring poor physical and mental health days compared to cisgender participants. Com-parison of results with 2015 data showed no differences in health outcomes or access, re-flecting the slow pace of change in improving care provision for transgender individuals (90).
Similar to our findings, health issues and needs that are of specific concern for transgender populations still remain unmet. Transgender people often pursue gender-affirming health care, such as hormone therapy and/or surgeries facing significant chal-lenges. In a 2023 systematic review of factors sociodemographic factors, treatment-related factors, psychosocial factors and, particularly health care interactions were identified as strong determinants of experiences transgender people report in regards to associated gender-affirming healthcare (91), Another recent study linked gender-affirmative care to the regular issue faced by transgender individuals who, in many countries, must undergo a psychosocial assessment and receive a letter of support from a mental health care pro-vider to access hormone and other specific treatment. In this study transgender partici-pants reported the increased psychological distress experienced to the psychosocial as-sessment as a significant barrier to gender-affirmative care (92).
Furthermore, transgender individuals face specific reproductive and sexual health needs which are also obstructed with significant barriers. In agreement with the barriers identi-fied by our review, a recently published meta synthesis of evidence on the fundamental problems that the transgender population faces regarding experiences of sexual and re-productive health illustrates the impact of individual, interpersonal, institutional, and so-cial factors that influence this health aspect, including limited information, lack of aware-ness, low socioeconomic status, stigma, discrimination, and social deprivation (93). An-other recent literature review further emphasizes that research on specialized reproductive healthcare for transgender individuals (e.g. medical interventions and fertility, gamete preservation, etc.) is still not receiving proper attention, while routine interventions such as preconception counseling, prenatal surveillance, perinatal support, contraceptive, and pregnancy termination related healthcare are still lacking meaningful adaptation for this patient population and many knowledge gaps remain to be filled (94).
To the same direction, barriers to care in regards to sexually transmitted diseases and, par-ticularly HIV, place specific challenges to the transgender population In a large 2023 USA study focusing on indices of HIV risk among transgender women, unexpectedly high mor-tality was found. Although none of these deaths was HIV-related, the authors suggested that the issue is indicative of poor care and should not be neglected (95). In terms of essen-tial Pre-Exposure Prophylaxis (PrEP) medication for HIV prevention, another recent scop-ing review found high willingness (80%) to use PrEP among transgender women world-wide, yet low uptake and adherence (35.4%), with socioeconomic determinants, including poverty, substance use, stigma and mistrust again driving this phenomenon (96), in line with the barriers identified by this study.
Other studies have also examined barriers in mental health care for transgender individu-als, another particularly challenging area for this population. One of the first systematic reviews to specifically explore obstacles to transgender mental health care identified major barriers which are similar to the ones also reported in this review and include personal concerns, involving fear of people of being pathologized or stereotyped, objection to com-mon therapeutic practices and incompetent mental health professionals, including those who are unknowledgeable, unnuanced, and unsupportive and affordability obstacles faced by transgender individuals (97).
In terms also of common chronic diseases, such as cancer, a 2023 narrative scoping re-view focusing on transgender cancer care identified similar barriers to our results suggest-ing suboptimal service delivery in the continuum of care. This review revealed that transgender were not only burdened by high prevalences of risk factors (e.g. tobacco, alco-hol) but also high rates HPV and HIV-related cancers. They were less likely to adhere to cancer screening, while socio-economic determinants seemed to drive these disparities. Lack of knowledgeable healthcare practitioners represented a major hurdle to cancer pre-vention, care, and survivorship for this population, while discrimination, discomfort caused by gender-labeled oncological services, stigma, and lack of cultural sensitivity of health care practitioners were other barriers met in the oncology setting (98).
In line with our results, system-level barriers have also been documented in more recent research. In a qualitative study of US transgender adults, misgendering and system-wide insensitivity during health care encounters and low levels of understanding of their transgender experience among primary care providers were mentioned as significant ad-versities. Provider-patient relationship improvements and service-provider sensitivity training. better care coordination and enhancing patient support for navigation of gender affirmation services, careful consideration when implementing systemwide routine pro-cesses such as using correct pronouns and names were recommended as ways to over-come these barriers (99).
Despite the limited evidence from Europe compared to the USA, results from a 2023 study evaluating experienced barriers to care for transgender individuals (n=307) in three coun-tries (Belgium, the Netherlands, Germany) may be indicative of the European landscape. In this study, the majority of participants reported various barriers, with the most-frequently reported being the lack of support from the social environment, travel time to clinics and costs and stiff treatment protocols in the sense that many participants reported they had to convince their provider they needed care and/or express their wish in such way to increase their likelihood of receiving care. More mental health problems and lower income and female gender were again associated with more negative experiences, sug-gesting that European health systems need to address both personal and systemic charac-teristics to overcome barriers to transgender care (100)”.
3) The focus of the paper (as observed from the title) is on Greece, will the authors consider giving a brief highlight on the cultural and political/social barriers for trans (if possible) or LGBT people to access health care in Greece specifically? in the introduction and the discussion sections?
Response: Thank you. In lack of robust evidence particularly for the trans community in Greece and, to the best of our knowledge, the information already provided in our introduction and discussion sections reflects the most recent political/social and cultural context of the country, which is prevailed by barriers relateted to absence of protective legal frameworks, denial of fundamental rights from substantial parts of the society due to religious and cultural beliefs, lack of inclusive health system and competent professionals and general fragmentation of the health system and lack of care integration. Namely, as already mentioned in our Introduction section: “Specifically for Greece, the problem is evident in all areas of life. Starting from the educational system, which is highly centralized and lacking sex education classes, attitudes towards LGBTQI+ people have been continously negative, although this is slowly changing. In a 2020 report from the International LGBT Association (ILGA) which assessed LGBTQI+ rights Europe, Greece was ranked 13th among 49 other European countries, with 48% improvement regarding legal and policy practices towards trans individuals in the thematic categories of quality and non-discrimination, family, hate crime and hate speech, legal gender recognition and bodily integrity, civil society space and asylum (9). However, this does not reflect the change in general attitudes of the public. The 2019 report by the Pew Research Center indicated that 48% of Greek respondents believed that society should accept homosexuality, while 47% did not believe so (10). Younger people (18-29 years old) were found more tolerant than older people (50 years and older) regarding diversity. Still, local particularities including the strong and persistent societal influence of religion and church, the financial crisis that hit the country the last decade and the rise of far-right political views have multiplied homophobic and transphobic incidents (9). One of the most important measures against homophobia and transphobia that Greek LGBTQI+ organisations are bringing forward is education offered in schools, along with targeted training of public officials including teachers, police officers, health care professionals and clinic staff etc. (11). Furthermore, Greek LGBTQI+ organisations strongly advocate for a complete ban of practices that aim to change gender identity and/or sexual orientation (known as conversion therapies)”.
As also mentioned in our Discussion section: “However, Greece faces substantial challenges in regards to meeting the above requirements. According to the Eurobarometer on Discrimination (2015) reports (85), Greece shows high rates of discrimination on sexual orientation (71%) and discrimination on gender identity (73%). The annual Greek Racist Violence Recording Network (RVRN) data from 2012 until 2017 shows high rates of verbal and/or physical violence, with 934 incidents of racist violence incidents concerning more than 1,000 victims and, more specifically, 98 violent incidents against trans people between 2015-2018 (86–88). In 2014, a law on hate crimes and hate speech was upvoted from the Greek parliament, prohibiting hate crimes and hate speech on the basis of gender identity (N. 4285/2014). Two years later the legal framework on equal treatment and combating discrimination (N. 4443/2016) was also updated to include a prohibition of all forms of discrimination on the basis of gender identity, but only in the sectors of employment, and accessing services and goods (89).According to Giannou (2017), trans people also seem to be excluded from healthcare due to negative stereotypes, stigma and general social assumptions (83). One of the most important steps regarding the protection of trans people’s rights in Greece was the introduction of a new law on legal gender recognition in 2017 (L. 4491/2017), which allowed trans people to change their name and gender on their official documents, without the requirement for psychiatric evaluation or other medical procedures (e.g, hormone therapy or gender-affirming surgeries). However, the law has several shortcomings, among which the costly and lengthy procedure trans people are required to follow, the lack of gender options outside the binary and the exclusion of people who are married, as well as underage persons under 17 years old. As a result, despite the current legal reform, not all trans people can (easily) access official documents that reflect their gender identity (89). As noted in the results, for trans people who wish to pursue gender-affirming medical interventions such as hormone-replacement therapy or surgeries, the high cost and general lack of specialised health professionals can be important obstacles. In Greece hormone-replacement therapy is covered by social health insurance, whereas all kinds of gender-affirming surgery are not. It is important to note that until ICD-11 is in enforced in Greece, trans people who want to access medical transition procedures are required to receive a psychiatric diagnosis of “Gender dysphoria” (90).Furthermore, problems and contextual issues that have been observed in the implementation of healthcare services in Greece, including the lack of integrated care and lack of patient-centred approaches could affect the establishment of effective doctor-patient relationships, therefore the delivery of appropriate care for transgender people by disregarding their healthcare needs (83,91). Primary care has been proven to provide a fruitful ground for designing and implementing novel approaches to enhance care for transgender individuals (92)”.
We hope this information sufficiently answers this comment and we thank the reviewer for the understanding.
Thank you for giving me the opportunity to read your paper.
Response: Thank you for your feedback.
Reviewer 2 Report
Comments and Suggestions for Authors
Based on a scoping review of 64 publications (located via a pubmed search since 2015) the authors conclude (lines 370-374): “Improving access to care for transgender people is a multidimensional issue that should tackle significant barriers at the personal, interpersonal and institutional level. Our review suggested that actions for future professional education initiatives should focus, among others, on respecting transgender identity, protecting confidentiality, creating trusted provider-patient relationships and providing sufficient competency on trans specific healthcare issues.”
Alhough the lack of quality of and access to (effective) health care for transgender and gender diverse persons is an important (research and clinical) topic, the current manuscript has important methodological and conceptual shortcomings:
(1). On line 86-90 the authors wrote: “The present review aims to synthesize the existing barriers that transgender individuals face in terms of access to healthcare globally and to discuss the potential gaps and needs for healthcare professionals’ training to improve transgender health in Greece”. Although the authors do not give a formal definition of healthcare, this reviewer assumes that the authors refer with healthcare to both so called somatic and mental health care. In line with this assumption, only searching in pubmed is methodologically not adequate enough: it introduces a bias in the selection of publications which could and should have been prevented by consulting other data bases (e.g. APA PsycInfo® ; scopus; ) too.
(2). Furthermore, the authors used as one of their inclusion criteria “Published from 2015 and onwards” (line 120). However, they did not justify why they used 2015 as a boarder (given their aim “to synthesize the existing barriers that transgender individuals face in terms of access to healthcare globally and to discuss the potential gaps and needs for healthcare professionals’ training to improve transgender health in Greece”).
(3). In the title of the manuscript “with a focus on Greece” is mentioned. Although this reviewer welcomes this specific focus on Greece, it remains unclear in the manuscript if their have been specific empirical studies in Greece on health care access/barriers for transgender and gender diverse persons. The authors wrote only (line 156-159): “The majority of identified research originated from the United States of America (USA) and featured qualitative methodology or survey data analysis. Very limited evidence was identified from Europe and much less from Greece”. Please describe/review the Greek empirical studies in detail, given “with a focus on Greece”! Now it remains unclear what has specifically been researched on access/barriers for transgender individuals in Greece.
(4).Furthermore, given the great differences in health care systems in the world (compare for example the differences between the health care systems in the United States of America, Scandinavian countries, Belgium, Russia…), the authors should make clear why they think that results of empirical studies done within different health care systems may be generalized to the Greek health care system.
(5). In the beginning of this manuscript we read: “ According to the World Health Organization (WHO), the enjoyment of the highest attainable standard of health is one of the fundamental rights of every human being. However, trans people worldwide experience substantial health disparities and barriers to appropriate healthcare services that keep them from achieving the highest possible health status”. Please add a definition of a health disparity and of a barrier. It is also given in consideration to elaborate shortly here (or in the results) how barriers and disparities were measured and what the validity is of the used measures.
Furthermore, there are also many minor problems:
(a).Please explain why you used the terminology “transgender individuals” and not “transgender and gender diverse individuals” (see also SOC-8 of the WPATH)
(b).In the abstract it is stated that 64 references were included in the review (line 19). However, in line 150 we read “Eventually, 69 were found relevant and included in the present analysis”. Clearly one of the numbers is wrong.
(c).Line 23: “ stereotypical system approaches”. What do you mean by “ stereotypical system approaches”. No were in the text, is this concept explained.
(d).Line 34-36: “However, trans people worldwide experience substantial health disparities and barriers to appropriate healthcare services that keep them from achieving the highest possible health status”. Insert reference of research (or review of empirical research) that substantiates this proposition.
(e).Line 41-44: “Prejudicial attitudes among health professionals and inherent heteronormativity in health services can deter Lesbian, Gay, Bisexual, Trans, Queer, Intersex and other persons (LGBTQI+) from seeking medical care.” Insert reference of research (or review of empirical research) that substantiates this proposition.
(f).Line 46-49 : “Although existing law implementing the principle of equal treatment between women and men is, to a certain extent, relevant to discrimination on grounds of gender identity, there is no legal framework when it comes to discrimination on the basis of sexual orientation in any area outside employment”. Formulated in this global way, this is incorrect. For example, in the Netherlands there exists such a formal legal framework. Please, be more nuanced!
(g). Line 53-60: I do not understand that research from 2013 contradicts the results of the 2020 report: things could have been changed between 2013 and 2020. Please clarify. Furthermore, I do not understand what you mean by “with 48% improvement regarding perceptions towards trans individuals”. Please clarify.
(h).Line 61-63: “Still, local particularities including the strong and persistent societal influence of religion and church, the financial crisis that hit the country the last decade and the rise of far right political views have multiplied homophobic and transphobic incidents.” On which research is this based? Add reference. If it is only a hypothesis or an opinion, please mention this.
(i). Line 84-85: “Since 2022, the use of ICD-11 is enforced in all countries”. Is it also used in all countries?
(j).Line 101-103: “As such, review of diverse sources including international, Greek and grey literature was implemented to explore and report on the existing situation and barriers prevailing health systems, focusing on the access of transgender individuals to healthcare services.” Skip this sentence, it adds no additional information.
(k).Line 122-123: “With the exception of literature reviews, reports, doctoral dissertations, or other documents that did not include primary data (e.g., protocols, conference abstracts etc.) were excluded.” If I am not mistaken, literature reviews and some other categories were included in your review, making it impossible (for the reader) to understand how many of the included studies were specific empirical projects. Please clarify. Compare with your abstract, in which you wrote
“Out of 560 identified references, 64 were included”, without any specification of the categories included (e.g. 60 empirical research projects and 4 literature reviews or 4 empirical research projects and 60 literature reviews).
(l).Line 176: “Some studies”. Please do not use some (or many), but mention the number of studies.
(m).Line 231: Please explain what you mean by a trans-specific service.
(n).Paragraph 4.3. Much of the information on the situation in Greece is new and should -in the view of this reviewer- be placed in earlier parts of the paper (e.g. in the introduction).
(0).In the discussion the authors suggests some possible interventions to improve the health care for transgender and gender diverse people. But they do not explain how these suggestions arise from the reported research; neither seems these suggestions be based on existing research on the effectivity of interventions to improve health care access and to delete health barriers and disparities. Please clarify/elaborate.
(p). The paragraph “ 4.4. Transgender health and Covid-19” does not belong in this paper. Delete it.
It is a entirely new topic that is introduced in the discussion and has no connection to the research question. If you view (the handling or consequences of) Covid-19 as a mediator or moderator of barriers/disparities you should pay attention to this variable/influence in the introduction and outline how you have researched this variable in your study.
In conclusion, the current manuscript is superficial and has a many probems, which need a thorough revision.
Author Response
Reviewer 2
Based on a scoping review of 64 publications (located via a pubmed search since 2015) the authors conclude (lines 370-374): “Improving access to care for transgender people is a multidimensional issue that should tackle significant barriers at the personal, interpersonal and institutional level. Our review suggested that actions for future professional education initiatives should focus, among others, on respecting transgender identity, protecting confidentiality, creating trusted provider-patient relationships and providing sufficient competency on trans specific healthcare issues.”
Although the lack of quality of and access to (effective) health care for transgender and gender diverse persons is an important (research and clinical) topic, the current manuscript has important methodological and conceptual shortcomings:
Response: We thank the reviewer for the valuable feedback and we hope our revised manuscript to have sufficiently addressed all raised concerns.
(1). On line 86-90 the authors wrote: “The present review aims to synthesize the existing barriers that transgender individuals face in terms of access to healthcare globally and to discuss the potential gaps and needs for healthcare professionals’ training to improve transgender health in Greece”. Although the authors do not give a formal definition of healthcare, this reviewer assumes that the authors refer with healthcare to both so called somatic and mental health care. In line with this assumption, only searching in pubmed is methodologically not adequate enough: it introduces a bias in the selection of publications which could and should have been prevented by consulting other data bases (e.g. APA PsycInfo®; scopus;) too.
Response: Thank you for this comment. Since this is a new research topic for the Greek medical context, our aim was to explore the wider evidence that has been published as a starting point of our own research activities. This is why we chose to specifically perform a scoping review (following the definition of Colquhoun HL, et al. J Clin Epidemiol. 2014) which is an exploratory rather than systematic approach of literature search. We already explain the definition and rationale of choosing this methodology in our Design subsection, where we mention “A scoping review was conducted to provide a synthesis of the evidence from diverse healthcare studies and help with informing clinical practice and health policy (13). Rather than evaluating or weighting the findings of individual studies, scoping reviews provide a snapshot of an overlooked or emergent field of research”. Additionally, since our focus was on the medical rather that phycological/social field (yet including both somatic and mental health), we indeed restricted our search in PubMed, one of the most acknowledged and acurate medical databases worldwide. We have already acknowledged that in our Strengths and Limitations subsection, were we already mention “Firstly, we have searched in only one database, potentially missing relevant papers that are not included in PubMed. Although we noticed the majority of our identified factors reoccurred throughout the papers (leading to data saturation), publications from other databases may potentially result in a slight shift in key themes”.
However, to make our point even more clear and address the reviewer’s comment, we added in our Search Strategy subsection of the Methods section the following sentence “The database of PubMed was used for the search of peer-reviewed articles and indexed publications (September – October 2020), since the focus of this issue was on the medical, rather than phycological/social field (yet without excluding either somatic or mental healthcare articles)”.
(2). Furthermore, the authors used as one of their inclusion criteria “Published from 2015 and onwards” (line 120). However, they did not justify why they used 2015 as a boarder (given their aim “to synthesize the existing barriers that transgender individuals face in terms of access to healthcare globally and to discuss the potential gaps and needs for healthcare professionals’ training to improve transgender health in Greece”).
Response: Thank you and we have applied this criterion in order to retrieve recent evidence from the last 10 years (until the review) as, on the one hand, research was limited in earlier years and, on the other hand, the many changes have been observed the latest years in the field of transgender health and rights in general. To address this comment, in our Search Strategy subsection of the Methods section we modified the third inclusion criterion to now read: “Published from 2015 and onwards, so as to retrieve recent evidence published within the last 10 years”
(3). In the title of the manuscript “with a focus on Greece” is mentioned. Although this reviewer welcomes this specific focus on Greece, it remains unclear in the manuscript if their have been specific empirical studies in Greece on health care access/barriers for transgender and gender diverse persons. The authors wrote only (line 156-159): “The majority of identified research originated from the United States of America (USA) and featured qualitative methodology or survey data analysis. Very limited evidence was identified from Europe and much less from Greece”. Please describe/review the Greek empirical studies in detail, given “with a focus on Greece”! Now it remains unclear what has specifically been researched on access/barriers for transgender individuals in Greece.
Response: We thank the reviewer for this comment. Indeed, almost no research article evidence was found from Greece, apart from one identified in grey literature, along with other general reports from LGBTQI+ organisations. We thus modified the respective sentence to now read “The majority of identified research originated from the United States of America (USA) and featured qualitative methodology or survey data analysis. Very limited evidence was identified from Europe. From Greece, documents were identified only in gray literature and mainly concerned general reports of LGBTQI+ organizations with only one report presenting research focused on understanding the health inequalities of LGBT people in the country”.
(4). Furthermore, given the great differences in health care systems in the world (compare for example the differences between the health care systems in the United States of America, Scandinavian countries, Belgium, Russia…), the authors should make clear why they think that results of empirical studies done within different health care systems may be generalized to the Greek health care system.
Response: We thank the reviewer for this comment. Our aim was not to generalize the results from other settings to the Greek healthcare system but, in absence of particular and solid evidence from Greece, we wanted to explore what is known internationally, in order to understand the situation and prepare for the next research actions of our project. To clarify that, we rephrased the last paragraph of our Introduction section which states the aim of our study to now read: “This study is part of the European collaborative project TRANSCARE which seeks to improve access to healthcare for transgender populations (https://transcare-project.eu/). In absence of particular and solid evidence from Greece, the present review aims to synthesize the existing barriers that transgender individuals face in terms of access to healthcare globally and to discuss how these results can be used to further explore and address the potential training gaps and needs of Greek healthcare professionals, so as to improve the quality of care for the local transgender population”.
(5). In the beginning of this manuscript we read: “According to the World Health Organization (WHO), the enjoyment of the highest attainable standard of health is one of the fundamental rights of every human being. However, trans people worldwide experience substantial health disparities and barriers to appropriate healthcare services that keep them from achieving the highest possible health status”. Please add a definition of a health disparity and of a barrier. It is also given in consideration to elaborate shortly here (or in the results) how barriers and disparities were measured and what the validity is of the used measures.
Response: Thank you. We have already provided several barriers in the immediately subsequent sentence of the one that is being quoted here. In particular, we already report that “Barriers to healthcare experienced by trans communities include discriminatory treatment by healthcare providers, a lack of providers who are trained to offer appropriate healthcare and refusal by many national health systems and health insurance programs to cover services for trans people (2)”. To further address the reviewer’s comment, we added in the first paragraph of our Introduction section a definition and examples (along with their respective reference) of health disparities as follows: “Health disparities as defined by differences in health outcomes among transgender com-pared to cisgender individuals have also been highlighted in the 2019 Behavioral Risk Factor Surveillance System report that documents higher rates of adverse mental, physical and behavioral health outcomes for transgender people. According to the report, 60% of transgender adults report having poor mental health at least one day in the past month compared with 37% of cisgender adults, while 54% of transgender adults report having had poor physical health at least one day in the past month compared with 36% of cis-gender adults (3)”.
Furthermore, there are also many minor problems:
(a). Please explain why you used the terminology “transgender individuals” and not “transgender and gender diverse individuals” (see also SOC-8 of the WPATH)
Response: We used that terminology as an umbrella term that is used to describe the full range of people whose gender identity and/or gender role do not conform to what is typically associated with their sex assigned at birth. We clarified that on our Search Strategy subsection of the Methods section where we added: “The term “transgender” (or “trans” or “transexual”) is used here as the umbrella term that describes the full range of people whose gender identity and/or gender role do not conform to what is typically associated with their sex assigned at birth”.
(b). In the abstract it is stated that 64 references were included in the review (line 19). However, in line 150 we read “Eventually, 69 were found relevant and included in the present analysis”. Clearly one of the numbers is wrong.
Response: Thank you for noting this which, most probably, was a typographical error that slipped our attention. The correct number is 69 and we fixed that in the abstract.
(c). Line 23: “stereotypical system approaches”. What do you mean by “stereotypical system approaches”. No were in the text, is this concept explained.
Response: In order to clarify this and avoid further confusion we rephrased the respective sentence to now read: “…and healthcare systems that do not take into account particular transgender health issues during care provision”.
(d). Line 34-36: “However, trans people worldwide experience substantial health disparities and barriers to appropriate healthcare services that keep them from achieving the highest possible health status”. Insert reference of research (or review of empirical research) that substantiates this proposition.
Response: Thank you for noticing that the number of reference no. 1 was missing here. We have now added it in the respective sentence.
(e). Line 41-44: “Prejudicial attitudes among health professionals and inherent heteronormativity in health services can deter Lesbian, Gay, Bisexual, Trans, Queer, Intersex and other persons (LGBTQI+) from seeking medical care.” Insert reference of research (or review of empirical research) that substantiates this proposition.
Response: Thank you and we inserted the respective reference with number 4.
(f). Line 46-49: “Although existing law implementing the principle of equal treatment between women and men is, to a certain extent, relevant to discrimination on grounds of gender identity, there is no legal framework when it comes to discrimination on the basis of sexual orientation in any area outside employment”. Formulated in this global way, this is incorrect. For example, in the Netherlands there exists such a formal legal framework. Please, be more nuanced!
Response: Thank you and we have elaborated on the sentence to now read: “…in many cases, there is no legal framework when it comes to discrimination on the basis of sexual orientation in any area outside employment (3-7)”.
(g). Line 53-60: I do not understand that research from 2013 contradicts the results of the 2020 report: things could have been changed between 2013 and 2020. Please clarify. Furthermore, I do not understand what you mean by “with 48% improvement regarding perceptions towards trans individuals”. Please clarify.
Response: We thank the reviewer for this important comment. We indeed provide the most updated results and reference of the Pew Research Center survey of 2019, which still confirm our statement. As such, we have modified the respective sentence of the Introduction section to now read: “However, this does not reflect the change in general attitudes of the public. The 2019 report by the Pew Research Center indicated that 48% of Greek respondents believed that society should accept homosexuality, while 47% did not believe so (10)”.
As far as the meaning of the sentence “…with 48% improvement regarding perceptions towards trans individuals” is concerned, this refers to improvements in the Rainbow Map and Index, ILGA-Europe’s annual benchmarking tool that illustrates the legal and policy situation of LGBTI people in Europe. We clarified what this entails by adding text to the respective sentence of the Introduction section which now reads: “…with 48% improvement regarding legal and policy practices towards trans individuals in the thematic categories of quality and non-discrimination; family; hate crime and hate speech; legal gender recognition and bodily integrity; civil society space; and asylum (9)”.
(h). Line 61-63: “Still, local particularities including the strong and persistent societal influence of religion and church, the financial crisis that hit the country the last decade and the rise of far right political views have multiplied homophobic and transphobic incidents.” On which research is this based? Add reference. If it is only a hypothesis or an opinion, please mention this.
Response: Thank you and we added the respective reference from ILGA-Europe (reference number 9).
(i). Line 84-85: “Since 2022, the use of ICD-11 is enforced in all countries”. Is it also used in all countries?
Response: According to the WHO 35 were using ICD-11 as of February 2022. We added that information along with its reference in the respective sentence of the Introduction section which now reads: “Since 2022, the use of ICD-11 is enforced in all countries, with 35 countries actually using it as of February 2022 according to the WHO (14)”.
(j). Line 101-103: “As such, review of diverse sources including international, Greek and grey literature was implemented to explore and report on the existing situation and barriers prevailing health systems, focusing on the access of transgender individuals to healthcare services.” Skip this sentence, it adds no additional information.
Response: Thank you and we have now deleted this sentence.
(k). Line 122-123: “With the exception of literature reviews, reports, doctoral dissertations, or other documents that did not include primary data (e.g., protocols, conference abstracts etc.) were excluded.” If I am not mistaken, literature reviews and some other categories were included in your review, making it impossible (for the reader) to understand how many of the included studies were specific empirical projects. Please clarify. Compare with your abstract, in which you wrote “Out of 560 identified references, 64 were included”, without any specification of the categories included (e.g. 60 empirical research projects and 4 literature reviews or 4 empirical research projects and 60 literature reviews).
Response: Thank you. In this sentence the word “or” was redundant and changed the meaning of this sentence. We deleted that to correct the meaning and the sentence now reads: “With the exception of literature reviews, reports, doctoral dissertations, other documents that did not include primary data (e.g., protocols, conference abstracts etc.) were excluded. Due to word count limitations, we did not change anything else in the abstract (apart from correcting number 64 to 69 as per the reviewer’s previous observation).
(l). Line 176: “Some studies”. Please do not use some (or many), but mention the number of studies.
Response: Thank you and we deleted the word “some”. The sentence now reads: “Studies also suggested…”
(m). Line 231: Please explain what you mean by a trans-specific service.
Response: The sentence has been modified to now read: “Apart from the limited access to general healthcare services, the lack of services specifically for transgender individuals and their particular health needs (e.g. mental, sexual, reproductive, endocrinological etc. health issues experienced specifically by transgender people)…”.
(n). Paragraph 4.3. Much of the information on the situation in Greece is new and should -in the view of this reviewer- be placed in earlier parts of the paper (e.g. in the introduction).
Response: We thank the reviewer for this comment. As mentioned in our aim, it was among our goals to discuss how this international evidence could be viewed in the Greek context (in absence of respective evidence and research from our country) to inform further actions towards improving transgender health. As such, we kindly believe that discussing our findings in the light of the Greek situation fits well in the discussion section where we try to understand what is could be possibly done and what would possibly face challenges in the local context. Our results repeatedly highlight that lack of adequately trained healthcare professionals is one of the most frequently reported barriers to care for transgender individuals. In Greece, transgender care is not part of any level of healthcare professionals’ training (under or post graduate or continuous medical education). As such, we elaborate our discussion with some more focus on the importance of training (paragraph 4.3), which was also a significant part of the activities of the overall TRANSCARE project. We hope to have clarified our rationale on maintaining our Discussion section as it is and we thank the reviewer for the understanding.
(o). In the discussion the authors suggests some possible interventions to improve the health care for transgender and gender diverse people. But they do not explain how these suggestions arise from the reported research; neither seems these suggestions be based on existing research on the effectivity of interventions to improve health care access and to delete health barriers and disparities. Please clarify/elaborate.
Response: We thank the reviewer for this comment. As also per reviewer’s 1 request, we have added extensive and updated references in our Discussion section to link recent evidence (published since 2021) to our results and we hope this also made clear how our proposals arise from our results as well as the results of international literature. To address this comment even more specifically, we also added text on the evidence-base of interventions proposed in literature where we comment on ther effectiveness potential, merely focusing on educational strategies, the need of which has been highlighted both by our review and the published literature. As such, in the last paragraph of our Discussion section, the following has been inserted: “The evidence on the development and implementation of interventions to tackle barriers to care for transgender individuals is growing in the last years, while ongoing studies will determine the efficacy and effectiveness of these interventions. In particular, a literature review focusing on HIV care for transgender individuals identified promising interventions that address structural and individual barriers including societal and cultural stigma and highlight the fact that such interventions are built on the basis of ensuring the meaningful participation of the trans community in their design and implementation, the development of programs that ensure the integration gender-affirming care and social services with HIV care, the focus on improving behavioral health outcomes, the deployment of peer-led counseling, education, and navigation and the provision of technology-based interventions to increase access to care (111). Another extensive literature review provides detailed information on intervention research conducted so far with the aim to reduce barriers to care for transgender individuals, highlighting the significant potential of interventions that focus raising healthcare professionals’ education and skills, including educational interventions in medical students and residents and continuous medical education workshops, courses, curricular additions, simulation activities and other exercises (112)”.
(p). The paragraph “ 4.4. Transgender health and Covid-19” does not belong in this paper. Delete it.
It is a entirely new topic that is introduced in the discussion and has no connection to the research question. If you view (the handling or consequences of) Covid-19 as a mediator or moderator of barriers/disparities you should pay attention to this variable/influence in the introduction and outline how you have researched this variable in your study.
Response: Thank you and we acted as suggested. Paragraph 4.4 has been now deleted.
In conclusion, the current manuscript is superficial and has a many problems, which need a thorough revision.
Response: We thank the reviewer for the constructive commentary. We have extensively revised our manuscript based on both reviewers’ reports and we hope that it now meets the publication standards.